# Fatty Acid Composition of Meat and Edible Offal from Free-Living Red Deer (*Cervus elaphus*)

**DOI:** 10.3390/foods9070923

**Published:** 2020-07-14

**Authors:** Violeta Razmaitė, Vidmantas Pileckas, Artūras Šiukščius, Violeta Juškienė

**Affiliations:** Department of Animal Breeding and Reproduction, Animal Science Institute, Lithuanian University of Health Sciences, R. Žebenkos 12, 82317 Baisogala, Lithuania; Vidmantas.Pileckas@lsmuni.lt (V.P.); Arturas.Siukscius@lsmuni.lt (A.Š.); Violeta.Juskiene@lsmuni.lt (V.J.)

**Keywords:** game, meat, fat, fatty acids, offal, red deer

## Abstract

The objective of the study was to characterize tissue-associated differences in the fatty acid composition of fat in skeletal muscles *M. longissimus dorsi* (loin), *M*. *biceps femoris* (hind quarter), and *M. triceps brachii* (shoulder), and internal organs (i.e., liver, heart, and kidney) from free-living red deer (*Cervus elaphus)* females (*n* = 11) hunted in Lithuania. Skeletal muscles were characterized by lower content of free fat compared with the offal. The highest percentage of saturated fatty acids was found in the liver fat, whereas the lowest percentage was in the heart. Red deer offal showed significantly lower and higher proportions of monounsaturated and polyunsaturated fatty acids compared to meat, respectively. Higher proportions of oleic fatty acid in the shoulder and hind quarter compared to the loin were the only significant differences between skeletal muscles. The lowest and the highest n-6 polyunsaturated/n-3 polyunsaturated fatty acids (n-6/n-3PUFA) ratio were found in the liver and heart, respectively. More favorable lower atherogenic index and higher hypocholesterolemic/hypercholesterolemic ratio found in the offal showed their high nutritional value, however, higher peroxidizability index indicated higher susceptibility to lipid peroxidation compared to skeletal muscles.

## 1. Introduction

Mankind suffered a lack of meat and highly appreciated it throughout history, and even nowadays, meat is a very important source of energy and protein in food culture [1]. Significant increase in global meat consumption over time has been reported and trends for further meat consumption growth is expected [2,3]. However, the per capita meat consumption in the last 20 years in many European countries has stabilized [4] and different countries are located in different phases of the nutrition transition [5].

Moreover, a large amount of food, including meat and its products, is lost or wasted in various production, processing, and consumption stages [4]. Food wastes are associated not only with primary animal resources and by-products but also with environmental aspects. In spite of the fact that the number of slaughtered animals has increased over time, the use of edible by-products for human consumption has declined [6,7]. Consumption patterns vary depending on factors such as culture, traditions, and need, therefore, other authors noted that the consumption of animal by-products has continued to grow [8]. Although the most valuable parts for food are muscles, internal organs recovered from slaughtered and hunted animals offer a range of foods which have good nutritive value and constitute part of the diet in different countries worldwide [9,10,11,12]. The usefulness of some types of internal organs depends on the animal species from which they are obtained [7,10].

Consumers’ attitudes towards healthiness, safety, and taste are the most important factors influencing consumption intentions [1,13] but there are signs that the healthiness of meat is gradually becoming more important for consumers than safety concerns relating to meat products [1]. The concept of a healthy diet has gone through many transformations over time and continues to do so [4]. There is concern that high intake of meat is associated with chronic diseases such as obesity, diabetes, cardiovascular disease, and cancers [14,15]. However, other authors [16,17] have reviewed that there is limited evidence regarding red meat intake and increased risk of diseases. The evidence of civilization diseases and discussions in literature increased the interest in game meat, which is characterized by distinctive taste, high nutritional values, low fat content, favorable n-6/n-3 fatty acid ratios, and a high mineral content [18,19,20,21,22]. Concerning the offal of game animals, the studies were mainly focused on the risk of contamination [23,24,25].

Red deer is one of the most widely distributed deer species, being found in various geographical areas of Europe, Asia, and North America [26,27,28]. Although results have been published regarding the fatty acid composition in red deer meat [29,30,31,32], these studies presented the data on farmed red deer. Only a limited number of studies are available on the composition and quality of meat of wild free-living red deer [33,34,35,36]. Despite different previous studies, information on the nutritional value of offal from red deer is scarce. Thus the aim of this study was to characterize the tissue-associated differences in the fatty acid composition of different intramuscular, hepatic, and kidney lipids from free-living red deer (*Cervus elaphus*) hunted in Lithuania.

## 2. Materials and Methods

### 2.1. Animals and Sampling

Free-living red deer used in this study were shot in accordance with the law on hunting of the Republic of Lithuania, law No IX-966 of 18th June, 2013. In this experiment, 11 free-living red deer females of 2–3 years of age were hunted in the forests of the central–northern part of Lithuania in the latitude of 55°44′ to 55°68′ N and in the longitude of 23°66′ to 23°84′ E during the autumn–early winter hunting season (October–December). The samples were excised from the three skeletal muscles of different carcass sites, *M*. *longissimus dorsi* (loin), *M. biceps femoris* (hind quarter), and *M. triceps brachii* (shoulder), and samples of heart, liver, and kidney of females with carcass weight ranging from 54 to 66 kg in a special area for dressing of hunted animals. All these 66 samples were provided by the local hunters in a 24 h period after red deer shooting. The samples for analytical determination were stored at −65 ± 2.5 °C until analysis.

### 2.2. Fat Content

Fat was determined [37] by the Soxhlet extraction method (method No 960.39; AOAC 1990). The content of fat was expressed as weight percentage in wet tissue.

### 2.3. Fatty Acid Profiles

The extraction of lipids for fatty acid analysis was performed with a mixture of two volumes of chloroform (Chromasolv Plus, Sigma-Aldrich, St Louis, USA, for high performance liquid chromatography (HPLC) containing 0.5–1.0% ethanol as stabilizer) and one volume of methanol as described by Folch et al. [38]. Methylation of the samples was performed using sodium methoxide: 5 mL of 25 wt % solution in methanol was added to the sample and stirred. After 1 h, 7 mL HCL, 6 mL hexane, and 2 mL H_2_O were added. The top layer was transferred into a new test tube and evaporated. Fatty acid methyl esters were prepared according to the procedure described by Christopherson and Glass [39]. The fatty acid methyl esters (FAMEs) were analyzed using a gas–liquid chromatograph (GC—2010 SHIMADZU, Shimadzu corporation, Kyoto, Japan) fitted with flame ionization detector. The separation of methyl esters of fatty acids was affected on the capillary column Rt 2560 (100 m × 0.25 mm × 0.2 μm; Restek, Bellefonte, PA, USA) by temperature programming from 160 °C to 240 °C. The column was operated at 160 °C for 5 min, then the temperature was increased to 180 °C at 2 °C/min for 13 min, and then to 230 °C at 6 °C/min and held for 28 min. The temperatures of the injector and detector were held, respectively, at 240 °C and 260 °C. The rate of flow of carrier gas (nitrogen) through the column was 0.79 mL/min. The peaks were identified by comparison with the retention times of the standard fatty acids methyl esters 37 Component FAME Mix, BAME Mix, Linoleic acid methyl isomer mix (Supelco, Bellafonte, PA, USA), and trans FAME MIX k 110 (Grace, Deerfield, IL, USA). The relative proportion of each fatty acid was expressed as the relative percentage of the sum of the total fatty acids using “Lab solutions LC/GC” software, version 5.71 for Shimadzu gas chromatograph workstations.

The samples were analyzed at least in duplicate for all analytes.

### 2.4. Lipid Quality Indices

Lipid quality indices (i.e., atherogenic index (AI) and thrombogenic index (TI)) were calculated according to Ulbricht and Southgate [40]. The hypocholesterolemic/hypercholesterolemic (h/H) ratio was calculated according to Santos-Silva et al. [41]. The peroxidizability index (PI) was determined according to Du et al. [42].

### 2.5. Statistical Analysis

The data were subjected to the analysis of variance in general linear (GLM) procedure in IBM SPSS Statistics 22 with Bonferroni tests to determine the significance of differences of means between different tissues (*M. longissimus dorsi, M. biceps femoris, M. triceps brachii*, heart, liver, and kidney). The differences were regarded as significant when *p* < 0.05.

## 3. Results and Discussion

### 3.1. Fat Content and Saturated Fatty Acids

Skeletal muscles were characterized by a lower percentage of fat (*p* < 0.001) compared with the offal (Table 1). The highest (*p* < 0.001) fat content of red deer was in liver samples, whereas the lowest mean value within the offal was determined in the heart. In the present study, the meat fat content of female red deer were lower compared to data reported previously on meat from both sexes [33,34,43]. However, leanness of the red deer meat in the present study was in agreement with the review of Ramanzin et al. [44]. All females used in this study were hunted after red deer rutting season in Lithuania. Zomborszky and Husvéth [45] indicated significant decrease of fat in male liver during rutting season whereas the effect on females remains unknown. The similar ranges of fat content to those found in this study for liver, kidney, and heart were also reported for domestic bovine [46,47].

The highest percentage of total saturated fatty acids (SFA) was found in the liver fat compared both to skeletal muscles and the heart and kidney (*p* < 0.001). There were no significant differences between the skeletal muscles in the proportions of both total SFA and individual saturated fatty acids (Table 1). Palmitic (C16:0) and stearic (C18:0) acids were found to be dominant saturated acids in the studied muscles and offal of the red deer. The proportion of C16:0 in the heart was lower compared to skeletal muscles in the loin and hind quarter (*p* < 0.01) and shoulder (*p* < 0.05), while the proportion of C18:0 in the heart was higher (*p* < 0.01) compared only to the loin. The highest proportion of C18:0 was found in the liver (*p* < 0.001) compared to all studied samples. The kidney also had higher proportions of C18:0 than the loin (*p* < 0.001) and hind quarter (*p* < 0.05). Predominance of these fatty acids in the skeletal muscles over other saturated fatty acids is in agreement with the data of Polak et al. [33], Strazdinia et al. [43], and Daszkiewicz and Mesinger [35]. Literature review showed that fat high in stearic acid is no longer considered to increase plasma LDL-cholesterol concentrations and TC/HDL-cholesterol ratio compared with cholesterol-raising saturated palmitic fatty acid [48]. Although in the present study skeletal muscles, except shoulder, had higher contents of C16:0, heart, liver, and kidney had higher contents of C18:0. Similar ranges of C18:0 percentages for bovine offal except kidney were found by Florek et al. [46] and Alfaia et al. [47]. Kidney of domestic cattle in contrast to red deer had lower percentage of C18:0 than of C16:0. The offal of red deer also had higher proportions of other minor saturated fatty acids, particularly behenic (C22:0) fatty acid.

### 3.2. Monounsaturated Fatty Acids

The percentages of total monounsaturated fatty acids (MUFA) in the skeletal muscles were similar to those (19.9) reported by Strazdinia et al. [43], however, Daszkiewicz and Mesinger [35] have reported higher (31.96) proportion of total MUFA compared with the data obtained in the present study (Table 2). Red deer offal showed significantly lower proportions (*p* < 0.001) of total MUFA compared to meat. Within the MUFA, oleic (C18:1n-9) fatty acid was found to be dominant in all the samples and this is in agreement with the data of other authors both in the muscles of red deer [32,35,36] and in the offal of other ruminants [46,47].

In the present study, oleic (C18:1n-9) fatty acid was followed by the cis vaccenic (C18:1n-7) and isomers of palmitoleic (C16:1n-9 and C16:1n-7) fatty acids. A similar trend was reported by Alfaia et al. [47] for bovine offal and Lorenzo et al. [32] for red deer meat. In the present study, higher proportions of C18:1n-9 in the shoulder and hind quarter (*p* < 0.05) compared to the loin were the only significant differences between the skeletal muscles. Many individual monounsaturated fatty acids (C18:1n-9, C14:1n-9, C16:1n-9, C16:1n-7, C17:1n-9) in the heart muscle showed the lowest values compared not only to skeletal muscles but also to the liver and kidney. The lowest percentages of oleic and palmitoleic fatty acids in the heart of suckler beef were found by Florek et al. [46]. Within the MUFA, elaidic (C18:1*trans*-9) and palmitelaidic (C16:1*trans*-9) *trans* fatty acids were found to be dominant in the liver fat.

### 3.3. Polyunsaturated Fatty Acids

The total proportions of polyunsaturated fatty acids (PUFA), including many individual fatty acids in the kidney, liver, and heart, were higher compared to the skeletal muscles (Table 3). Linoleic (C18:2n-6) fatty acid was the dominant polyunsaturated fatty acid in all the muscles followed by arachidonic (C20:4n-6), α-linolenic (C18:3n-3), DPA (C22:5n-3), and EPA (C20:5n-3) fatty acids, whereas arachidonic (C20:4n-6) fatty acid was found to be dominant in the liver and kidney followed by C22:5n-3, C18:2n-6, and DHA (C22:6n-3) in the liver and by C18:2n-6, C22:5n-3, and C20:5n-3 fatty acids in the kidney.

There is important evidence relating to the beneficial and protective effects of n-3 omega PUFAs on consumer health. Although dietary α-linolenic acid can be converted to the EPA and DHA [49], other recent reports have also demonstrated that humans hardly convert α-linolenic acid into EPA and DHA and the conversion rate is very slow, thereby making EPA and DHA regarded as very essential [50]. Skeletal muscles in the present study had 20.3–35.5% and 60.6–80.8% higher contents (*p* < 0.001) of α-linolenic fatty acid than the heart and liver, respectively. EPA+DHA contents (*p* < 0.001) in the liver were 91.3% higher than in the loin to 2.4 times higher than in the hind quarter. In the kidney, the sum of these fatty acids was 94% higher than in the hind quarter and 2.43 times higher than in the loin. Similarly, higher proportions (*p* < 0.001) of DPA and arachidonic fatty acids were found in the studied offal compared to skeletal muscles.

### 3.4. Trans Fatty Acids

The heart and kidney of red deer had lower percentages (*p* < 0.05) of *trans* fatty acids than meat (Table 4). Purchas et al. [51] have found a slightly higher percentage of total *trans* fatty acids in the offal than in the meat of pastured domestic ruminants. In the present study, this could be in agreement only in comparison of red deer skeletal muscles and liver.

Numerous studies showed that higher *trans* fatty acid intake was associated with a higher risk of coronary heart disease, however, due to quite low (1–8%) content of *trans* fatty acids in ruminant products, the studies mostly focus on the effects of industrial or total *trans* fatty acid intake and its reduction [52,53]. Other authors [54] have reported that all *trans* fatty acids gram for gram have largely the same effect on blood lipoproteins. The highest proportion of total *trans* fatty acids in the present study was found in the liver, however, it was lower than that reported by Purchas et al. [51] for meat and offal of domestic ruminants (beef and lamb). Therefore, the presence of trans fatty acids at a 1.43–3.12% level in the red deer muscles and offal should not be a limiting factor for the use of either meat or offal of red deer.

### 3.5. Ratios of Fatty Acids and Lipid Quality Indices

The red deer showed the PUFA/SFA ratio significantly above the minimum (0.4) recommended for the diet [55] in all the studied tissues (Table 4). The highest ratio was found in the kidney and it was higher (*p* < 0.001) compared to the skeletal muscles and the liver (*p* < 0.01). A similar trend was observed by Florek et al. [46] and Alfaia et al. [47] between bovine kidney and liver, however, the values for the PUFA/SFA ratio of domestic ruminants were lower and less favorable than for the red deer. In the present study, the PUFA/SFA ratio in the heart was higher (*p* < 0.05) than in the shoulder and hind quarter muscles. The tissue type also appeared to affect the n-6/n-3 PUFA ratio and some lipid quality indices by a different level of significance. The lowest n-6/n-3 PUFA ratio was found in the liver and the highest in the heart and this is in agreement with the data reported by Alfaia et al. [47]. A review by Wood et al. [55] and also recommendations of Bellagio’s report on healthy agriculture, healthy nutrition, and healthy people [56] emphasized that the n-6/n-3 PUFA ratio for a healthy diet should not exceed 4. It can be observed that the n-6/n-3 ratio in all the tissues was lower than recommended for the diet and lower not only compared with the values for the ratio n-6/n-3 in the offal of domestic bovine [46,47] but also in the meat of Iberian wild red deer [32]. As PUFA/SFA and n-6/n-3 ratios do not contain monounsaturated fatty acids, thus eliminating the effects of MUFA, the lipid quality indices were calculated. More favorable lower atherogenic (AI) indexes (*p* < 0.01) and higher hypocholesterolemic/hypercholesterolemic (h/H) ratios (*p* < 0.01) were found in the offal compared to skeletal muscles and showed high nutritional value of red deer offal. However, higher (*p* < 0.001) peroxidizability index (PI) in the offal showed higher susceptibility to lipid peroxidation compared to skeletal muscles.

## 4. Conclusions

Skeletal muscles were characterized by lower content of free fat compared with the offal. The fatty acid composition was significantly affected by the tissue type. The highest percentage of SFA was found in the liver fat, whereas the lowest percentage was found in the heart. Red deer offal showed significantly lower and higher proportions of MUFA and PUFA compared to meat, respectively. Higher proportions of oleic fatty acid in the shoulder and hind quarter compared to the loin were the only significant differences between skeletal muscles. The lowest n-6/n-3 PUFA ratio was found in the liver and the highest in the heart. PUFA/SFA and n-6/n-3 ratios in all the studied tissues of the red deer were significantly more favorable than recommended for the diet. More favorable lower atherogenic index and higher hypocholesterolemic/hypercholesterolemic ratio found in the offal showed their high nutritional value, however, higher peroxidizability index indicated higher susceptibility to lipid peroxidation compared to skeletal muscles. On the basis of the results obtained from the analysis of fatty acid composition, it could be concluded that the edible offal of red deer are suitable for incorporation in the human diet. Furthermore, the fatty acid ratios and indices of lipid quality both in skeletal muscles and edible offal of red deer could improve overall diet in relation to healthy nutrition.

## Figures and Tables

**Table 1 foods-09-00923-t001:** Effect of tissue on fat content and saturated fatty acid (% of total FA) composition in lipids of red deer.

Fatty Acids	Skeletal Muscles			Offal			SE	*p*-Value
	Loin (*M. longissimus dorsi)*	Hind Quarter (*M. biceps femoris)*	Shoulder (*M**. triceps brachii)*	Liver	Heart	Kidney		
Fat	0.65 ^a^	0.86 ^b^	0.88 ^b^	2.78 ^c^	1.29 ^d^	1.90 ^e^	0.043	<0.001
C12:0	0.03	0.06	0.04	0.05	0.04	0.06	0.027	0.969
C13:0	0.00 ^a^	0.00 ^a^	0.00 ^a^	0.00 ^a^	0.00 ^a^	0.09 ^b^	0.013	<0.001
C14:0	2.23 ^a^	2.49 ^a^	2.27 ^a^	0.84 ^b^	1.01 ^b^	0.45 ^b^	0.226	<0.001
C15:0	0.46 ^a^	0.57	0.61	0.79 ^b^	0.51 ^a^	0.44 ^a^	0.055	<0.001
C16:0	16.19 ^a^	16.33 ^a^	15.25 ^a^	13.27	10.63 ^b^	14.04	1.052	0.002
C17:0	0.53 ^a^	0.60 ^a^	0.58 ^a^	1.57 ^b^	0.46 ^a^	0.93 ^b^	0.050	<0.001
C18:0	13.73 ^a^	14.78 ^a’^	15.38 ^a’^	27.27 ^b,b’^	15.98 ^b,c^	16.86 ^b,b’^	0.415	<0.001
C20:0	0.04 ^a^	0.06 ^a^	0.06 ^a^	0.06 ^a^	0.09 ^a^	0.21 ^b^	0.019	<0.001
C21:0	0.21	0.28	0.27	0.25	0.16	0.13	0.038	0.051
C22:0	0.16 ^a^	0.15 ^a^	0.16 ^a^	0.51 ^b^	0.27 ^c^	0.37 ^d^	0.022	<0.001
SFA	33.56 ^a^	35.32 ^a^	34.62 ^a^	44.61 ^b^	29.15 ^a^	33.57 ^a^	1.524	<0.001

The differences between the means in the rows with different small letters with and without superscripts are significant; SE = pooled standard errors; SFA = sum of all identified saturated fatty acids.

**Table 2 foods-09-00923-t002:** Effect of tissue on monounsaturated fatty acid (% of total FA) composition in lipids of red deer.

Fatty Acids	Skeletal Muscles			Offal			SE	*p*-Value
	Loin (*M.* *longissimus dorsi*)	Hind Quarter (*M. biceps femoris)*	Shoulder (*M.* *triceps brachii)*	Liver	Heart	Kidney		
C14:1n-9	0.45 ^a^	0.58 ^a’^	0.45 ^a^	0.29 ^b’^	0.22 ^bb’^	0.33 ^b’^	0.047	<0.001
C15:1	0.59 ^a^	0.53 ^a^	0.55 ^a^	0.00 ^b,b’^	0.72 ^b,a’^	0.41 ^ba’^	0.027	<0.001
C16:1*trans*-9	0.43 ^a^	0.48 ^a^	0.47 ^a^	0.79 ^b^	0.49 ^a^	0.55 ^a^	0.054	<0.001
C16:1n-9	2.28	2.35	1.95	0.67	0.62	0.70	0.407	0.027
C16:1n-7	1.12	1.66 ^a^	1.37	1.18	0.58 ^b^	0.76	0.238	0.002
C17:1n-9	0.18 ^a^	0.22 ^a’^	0.20	0.17 ^a^	0.16 ^ab’^	0.25 ^b^	0.013	<0.001
C18:1*trans*-9	1.02	1.28	1.28	1.54 ^a^	0.88 ^b^	0.74 ^b^	0. 139	0.001
C18:1n-9	9.78 ^a^	12.23 ^b^	11.14 ^a^	7.45 ^b^	5.67 ^b^	10.45 ^a^	0.532	<0.001
C18:1n-7	2.88 ^a,a’^	2.53 ^a^	2.45 ^a^	1.61 ^b^	1.88 ^b,b’^	2.21	0.193	<0.001
C20:1n-9	0.04 ^a^	0.00 ^a^	0.01 ^a^	0.06 ^a^	0.00 ^a^	0.15 ^b^	0.014	<0.001
MUFA	19.00 ^a^	22.02 ^a,a’^	20.09 ^a^	13.95 ^b^	11.34 ^b,a’^	16.95 ^b’^	0.818	<0.001

The differences between the means in the rows with different small letters with and without superscripts are significant; SE = pooled standard error; MUFA = sum of all identified monounsaturated fatty acids.

**Table 3 foods-09-00923-t003:** Effect of tissue on polyunsaturated fatty acid (% of total fatty acid composition in lipids of red deer.

Fatty Acids	Skeletal Muscles			Offal			SE	*p*-Value
	Loin (*M*. *longissimus dorsi*)	Hind Quarter (*M. biceps femoris)*	Shoulder (*M**. triceps brachii)*	Liver	Heart	Kidney		
C18:2n-6t9,12	0.13 ^a^	0.16 ^a’^	0.16 ^a’^	0.44 ^b,b’^	0.15 ^a^	0.08 ^b’^	0.016	<0.001
C18:2n-6c9,t12	0.17 ^a^	0.18 ^a’^	0.15 ^a^	0.24^b^	0.13 ^a^	0.00 ^b,b’^	0.016	<0.001
C18:2n-6t9,c12	0.16	0.23 ^a^	0.17	0.10	0.12	0.07^b^	0.031	0.012
C18:2n-6	15.12 ^a^	13.97 ^a’^	14.17 ^a’^	7.66 ^b,b’^	17.06 ^b’^	14.43 ^a,a’^	0.565	<0.001
C18:3n-6	0.09	0.05	0.05	0.08	0.05	0.00	0.020	0.069
C18:3n-3	3.40 ^a^	3.29 ^a^	3.02 ^a’^	1.88 ^b,b’^	2.51 ^b,a’^	2.22 ^b,b’^	0.138	<0.001
C20:2n-6	0.00 ^a^	0.00 ^a^	0.00 ^a^	0.25 ^b^	0.04 ^a^	0.30 ^b^	0.015	<0.001
C20:3n-6	1.03 ^a^	0.89 ^a^	0.96 ^a^	2.02 ^b^	1.19 ^a^	1.11^a^	0.095	<0.001
C20:3n-3	0.35	0.30	0.25 ^a^	0.40	0.27 ^a^	0.66 ^b^	0.086	0.016
C20:4n-6	7.91 ^a^	7.15 ^a’^	8.07 ^a^	10.49 ^b’,a’^	13.72 ^b,b’^	15.41 ^b,a’^	0.655	<0.001
C20:5n-3	2.03	1.56 ^a^	1.55 ^a^	2.02	2.06	2.51^b^	0.145	<0.001
C22:2n-6	0.21 ^a^	0.17 ^a’^	0.15 ^a’^	0.49 ^b^	0.15 ^a,b’^	0.35 ^b,b’^	0.040	<0.001
C22:4n-6	0.34 ^a^	0.25 ^a^	0.28 ^a^	1.59 ^b^	0.30 ^a^	0.51 ^a^	0.144	<0.001
C22:5n-3	2.90 ^a^	2.60 ^a,a’^	2.82 ^a^	8.09 ^b^	3.14 ^a^	3.53 ^ab’^	0.210	<0.001
C22:6n-3	0.50 ^a^	0.42^a^	0.51 ^a^	2.82 ^b^	0.80 ^a^	1.35 ^b^	0.119	<0.001
EPA+DHA	2.53 ^a^	1.99 ^b’^	2.06 ^b’^	4.84 ^b,a’^	2.85 ^a^	3.86 ^b,a’^	0.162	<0.001
PUFA	34.31 ^a^	31.22 ^a,a’^	32.32 ^a^	38.56 ^b’^	41.68 ^b^	42.52 ^b^	1.480	<0.001

The differences between the means in the rows with different letters with and without superscripts are significant; PUFA = sum of all identified polyunsaturated fatty acids. EPA+DHA PUFA = sum of eicosapentaenoic (C20:5n-3) and docosahexaenoic (C22:6n-3) fatty acids.

**Table 4 foods-09-00923-t004:** Total trans fatty acids and fatty acid ratios and lipid quality indices in lipids from different red deer tissues.

Variables	Skeletal Muscles			Offal			SE	*p*-Value
	Loin (*M*. *longissimus dorsi*)	Hind Quarter (*M. biceps femoris)*	Shoulder (*M**. triceps brachii)*	Liver	Heart	Kidney		
TFA	1.90 ^a^	2.34 ^a’^	2.23 ^a^	3.12 ^b^	1.77 ^a^	1.43 ^a,b’^	0.202	<0.001
PUFA/SFA	1.09 ^a^	0.89 ^a,a’^	1.01 ^a,a’^	0.87 ^a^	1.51 ^b’^	1.27 ^b^	0.085	<0.001
n-6/n-3	2.75 ^a^	2.84 ^a’^	2.95 ^a’^	1.55 ^b,b’^	3.80 ^b,b’^	3.16 ^b,a’^	0.088	<0.001
AI	0.49 ^a^	0.50 ^a’^	0.48 ^a^	0.32	0.28 ^b’^	0.27 ^b,b’^	0.048	<0.001
TI	0.68	0.71	0.74	0.64	0.58	0.56	0.057	0.189
h/H	2.83 ^a^	2.46 ^a,a’^	2.93 ^a^	3.31 ^a^	4.68 ^b^	3.78 ^b’^	0.269	<0.001
PI	92.54 ^a^	82.26 ^a^	87.61 ^a^	149.77 ^b,b’^	119.59 ^b,a’^	134.36 ^b^	4.942	<0.001
UFA	13.35 ^a^	10.88 ^a^	13.18 ^a^	2.77 ^b,a’^	17.95 ^b,b’^	7.27 ^b,b’,a’^	0.780	<0.001

The differences between the means in the rows with different small letters with and without superscripts are significant; TFA = sum of all identified *trans* fatty acids; PUFA/SFA = ratio of ΣPUFA to ΣSFA, n-6/n-3 = ratio of Σn-6 PUFA to Σn-3 PUFA, AI = atherogenic index, TI = thrombogenic index, h/H = hypocholesterolemic/hypercholesterolemic ratio, PI = peroxidizability index. UFA = sum of unidentified fatty acids and their isomers.

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
