# Peer review of "Fatty Acid Composition of Meat and Edible Offal from Free-Living Red Deer (Cervus elaphus)"

_foods, 2020, doi:10.3390/foods9070923_

Round 1

Reviewer 1 Report

The authors presented interesting results regarding the fatty acid composition of three different skeletal muscles (longissimus dorsi, biceps femoris and triceps brachii) and internal organs (liver, heart and kidney) from free-living red deer. Before being considered for publication, the authors would need to address some editorial issues along with improving some parts of the manuscript that are not enough or not accurate.

Specific comments

Lines 9-10: this statement is too general, please delete.

Line 17: delete “fatty”.

Lines 19-22: please revise to improve readability.

Line 26: delete “meat”.

Line 67: please provide some more background information, if possible, of the carcass weights or some other description of animals.

Line 71: please describe how the samples were stored before each analytical determination.

Line 76: What is the methodology used for the extraction of lipids? Folch et al. (1957)? Please add some more details on lipid extraction and methylation.

What was the number of replicates for meat/offal fatty acid analysis?

Lines 105-107: suggested change “In the present study, the meat fat contents of female red deer were lower compared to data reported previously on meat from both sexes (Polak et al………………)”

Lines 127-132: move these sentences at line 122 before the discussion.

Lines 134-137; 177-179: same here… please rewrite more clearly. Reformulate these sentences in order to show your results and after the discussion with other studies. However, the results and discussion section should be improved.

Line 138: suggested change “Among the MUFA, oleic acid…”.

Line 155-157: “a different level of significance”??? please rewrite more clearly.

Line 187: delete “observational”.

Author Response

Response to Reviewer 1 Comments

Point 1: Moderate English changes required 

The text was edited by an English language specialist

Point 2: Lines 9-10: this statement is too general, please delete.

The comment was accepted and the sentence was deleted

Line 17: delete “fatty”.

Deleted

Lines 19-22: please revise to improve readability.

Tried to improve

Line 26: delete “meat”.

Deleted

Line 67: please provide some more background information, if possible, of the carcass weights or some other description of animals.

The carcass weight was added (Lines 70-71 in revised manuscript) 

 Line 71: please describe how the samples were stored before each analytical determination.

Description of sample storage was added (Lines 72-73 in revised version)

Line 76: What is the methodology used for the extraction of lipids? Folch et al. (1957)? Please add some more details on lipid extraction and methylation.

Details in description on lipid extraction and methylation were added. References to Folch et al[38] and Christopherson and Glass [39] were included (Lines 78-84 in revised text and Lines 357-360 in References)

What was the number of replicates for meat/offal fatty acid analysis?

The samples were analyzed at least in duplicate for all analytes. (Line 96 )

Lines 105-107: suggested change “In the present study, the meat fat contents of female red deer were lower compared to data reported previously on meat from both sexes (Polak et al………………)”

Revised according to suggestion (Lines113-115 in revised manuscript)

Lines 127-132: move these sentences at line 122 before the discussion.

It was revised (Lines 130-134 in revised manuscript)

Lines 134-137; 177-179: same here… please rewrite more clearly. Reformulate these sentences in order to show your results and after the discussion with other studies. However, the results and discussion section should be improved.

Revised according to suggestion (Lines 150 and 195-197 in revised text)

Line 138: suggested change “Among the MUFA, oleic acid…”.

We would like to leave „Within the MUFA“

Line 155-157: “a different level of significance”??? please rewrite more clearly.

Corrected (Line 173)

Line 187: delete “observational”.

Deleted (Line 208 in revised manuscript)

Reviewer 2 Report

Foods Manuscript 839407 “Fatty acid composition of meat and edible offal from free living red deer (Cervus elaphus)

General comments

The manuscript aimed to analyse and characterise differences in the fatty acid composition of fat from muscles and offal of hunted red deer. The topic addressed is interesting even if it is not completely original. Particularly, results about muscles are already present in literature, while results about offal are sporadic and incomplete.

The experiment is correctly designed and the paper is well written, even if the number of animals is quite low.

Tables have to be improved: the use of statistical symbols it is not clear and confusing. More details in “Specific comments” section.

The autocitation is appropriate.

Specific comments

-Be consistent in using ise/ize spelling (see for example line 10 and line 7).

-Line 31: exactly? What does it mean?

-Line 57: it’ true, data about offal are scarce. Anyway, you could cite and use for some comparison https://doi.org/10.5194/aab-62-227-2019, for example, even if the paper refers to farmed red deer. Also Zomborszky (not recent, 2000) (doi: 10.1016/s1095-6433(00)00195-1).

-Line 73: please, add AOAC in (method No 960.39).

-Line 78: please, add fatty acids methyl esters before FAMEs.

-Lines 97-100: “between (among) the groups”: please, better specify the groups considered in the statistical comparisons

-Line 99: not Bonferoni but Bonferroni.

-Table 1 and others: The use of many letters for statistical differences is confusing and not immediately understandable. If, as I guess, letters in each row are the result of all possible comparisons among the six groups, it is better to choose a single significance level and use different letters in each row to highlight statistical differences.

-Lines 105-109: the authors should better discuss the reasons why the fat content of muscle in their study is lower than that reported by other researchers, since many factors can have a role.

-Line 173: 2.65: please, check this value.

-Lines 211-212: “As PUFA/SFA and n-6/n-3 ratios eliminate the effects of MUFA”: please, explain more clearly this statement.

Author Response

Response to Reviewer 2 Comments

Point 1: English language and style are fine/minor spell check required 

The text was edited by an  English language specialist

Point 2:The experiment is correctly designed and the paper is well written, even if the number of animals is quite low.

The licences are strictly limited for hunter groups for red deer hunting in Lithuania, therefore we could not collect more samples during one season.

 Point 3: Tables have to be improved: the use of statistical symbols it is not clear and confusing. More details in “Specific comments” section

Specific comments

-Be consistent in using ise/ize spelling (see for example line 10 and line 7).

Tried to use "ize"(characterize) as in The English Illustrated  Dictionary (Oxford,1996)

-Line 31: exactly? What does it mean?

As irrelevant it was deleted

-Line 57: it’ true, data about offal are scarce. Anyway, you could cite and use for some comparison https://doi.org/10.5194/aab-62-227-2019, for example, even if the paper refers to farmed red deer. Also Zomborszky (not recent, 2000) (doi: 10.1016/s1095-6433(00)00195-1).

Suggested literature sources  were used in lines 57 and 156 (Nagy et al.[36], in lines 117-120 (Zomborszky and Husveth [45] and included in References

-Line 73: please, add AOAC in (method No 960.39).

Added (Line 75 in revised manuscript)

-Line 78: please, add fatty acids methyl esters before FAMEs.

Description of fatty acid methyl esters of the lipids preparation was included (Lines 83-84)

-Lines 97-100: “between (among) the groups”: please, better specify the groups considered in the statistical comparisons

The groups of all tissues were specified (Line 107)

-Line 99: not Bonferoni but Bonferroni.

Corrected

-Table 1 and others: The use of many letters for statistical differences is confusing and not immediately understandable. If, as I guess, letters in each row are the result of all possible comparisons among the six groups, it is better to choose a single significance level and use different letters in each row to highlight statistical differences.

Different superscript letters in each row show all comparisons among all the six groups. With the aim to follow the suggestion to choose a single significance level, we revised the indication of significance in the tables.

-Lines 105-109: the authors should better discuss the reasons why the fat content of muscle in their study is lower than that reported by other researchers, since many factors can have a role

Females used in this study were shot after rutting season,which lasts from the end of August until October in Lithuania,  hovewer, the rut symptoms for hunted females were not evaluated. Zomborszky and Husvéth (2000) evaluated the effect of rutting only on males.  Reference to Zomborszky and Husvéth was included in Lines 117-120

-Line 173: 2.65: please, check this value.

It was checked and corrected (2.43) (Line 191)

-Lines 211-212: “As PUFA/SFA and n-6/n-3 ratios eliminate the effects of MUFA”: please, explain more clearly this statement.

Tried to explain (Lines 232-233 in revised manuscript)

Reviewer 3 Report

The manuscript intitled “Fatty Acid Composition of Meat and Edible Offal from Free Living Red Deer (Cervus elaphus)” aimed to characterize the lipid content and composition in different organs of red deer within the 24 hours after slaughtering. The objective of the research work is to characterize the lipids from different organs (muscle, liver, heart, kidney) of red deer from hunting and try to establish a link with the nutritional qualities of offals.

My main comment is that the results are very descriptive. The study does not aim to elucidate a biological mechanism.

However, the experimental design is clear and methods well adapted to the answer the scientific objective.

Moreover the manuscript is well written and the original results well presented can serve the scientific community to access objective data on lipids in different organs of the red deer.

Lines 134-137 : The sentence is not clear. I suggest to remove “Although” at the beginning of the sentence

Line 2018: What do you mean by “free fat” ?

I have not seen the reference of the supplementary material in the manuscript. I do not understand this supplementary material. Please explain. 

Author Response

Response to Reviewer 3 Comments

Point 1: English language and style are fine/minor spell check required 

The manuscript was edited by an English language specialist

Point 2: The manuscript intitled “Fatty Acid Composition of Meat and Edible Offal from Free Living Red Deer (Cervus elaphus)” aimed to characterize the lipid content and composition in different organs of red deer within the 24 hours after slaughtering. The objective of the research work is to characterize the lipids from different organs (muscle, liver, heart, kidney) of red deer from hunting and try to establish a link with the nutritional qualities of offals.

My main comment is that the results are very descriptive. The study does not aim to elucidate a biological mechanism.

We agree that our results are descriptive. We had no purpose to investigate the biological mechanism of fatty acids formation in red deer tissues

Lines 134-137 : The sentence is not clear. I suggest to remove “Although” at the beginning of the sentence

Revised                                                                                                                             

Line 2018: What do you mean by “free fat” ?

It means that Soxhlet extraction was without hydrolysis. We guess that it better to use only „fat“

I have not seen the reference of the supplementary material in the manuscript. I do not understand this supplementary material. Please explain. 

We did not supply this supplementary material, it is most probably a mistake

Round 2

Reviewer 1 Report

The manuscript is well improved after revision.